# uafR: An R package that automates mass spectrometry data processing

**Chase A. Stratton[1,2], Yvonne Thompson[1], Konilo Zio [3], William R. Morrison, III[4], Ebony G. Murrell [1]** *

1 The Land Institute, Salina, KS, United States of America, 2 Department of Biology, Delaware State University, Dover, DE, United States of America, 3 Wooclap SA, Etterbeek, Belgium, 4 USDA-ARS, Agricultural Research Service, Center for Grain and Animal Health Research, Manhattan, KS, United States of America

* murrell@landinstitute.org

## Abstract

Chemical information has become increasingly ubiquitous and has outstripped the pace of analysis and interpretation. We have developed an R package, *uafR*, that automates a grueling retrieval process for gas -chromatography coupled mass spectrometry (GC -MS) data and allows anyone interested in chemical comparisons to quickly perform advanced structural similarity matches. Our streamlined cheminformatics workflows allow anyone with basic experience in R to pull out component areas for tentative compound identifications using the best published understanding of molecules across samples (pubchem.gov). Interpretations can now be done at a fraction of the time, cost, and effort it would typically take using a standard chemical ecology data analysis pipeline. The package was tested in two experimental contexts: (1) A dataset of purified internal standards, which showed our algorithms correctly identified the known compounds with $R^2$ values ranging from 0.827–0.999 along concentrations ranging from $1 \times 10^{-5}$ to $1 \times 10^3$ ng/µl, (2) A large, previously published dataset, where the number and types of compounds identified were comparable (or identical) to those identified with the traditional manual peak annotation process, and NMDS analysis of the compounds produced the same pattern of significance as in the original study. Both the speed and accuracy of GC -MS data processing are drastically improved with *uafR* because it allows users to fluidly interact with their experiment following tentative library identifications [i.e. after the *m/z* spectra have been matched against an installed chemical fragmentation database (e.g. NIST)]. Use of *uafR* will allow larger datasets to be collected and systematically interpreted quickly. Furthermore, the functions of *uafR* could allow backlogs of previously collected and annotated data to be processed by new personnel or students as they are being trained. This is critical as we enter the era of exposomics, metabolomics, volatilomes, and landscape level, high-throughput chemotyping. This package was developed to advance collective understanding of chemical data and is applicable to any research that benefits from GC -MS analysis. It can be downloaded for free along with sample datasets from Github at github.org/castratton/uafR or installed directly from R or RStudio using the developer tools: 'devtools::install_github("castratton/uafR")'.

**Data Availability Statement:** Datasets used in this analysis are available on GitHub: github.org/castratton/uafR.

**Funding:** This project was funded by USDA-NIFA projects: #2021-67034-35135 and #2018-67013-

27402; and by generous private donations to The Land Institute. In addition, this work was funded, in part, by a United States Department of Agriculture, National Institute of Food and Agriculture, Crop Protection and Pest Management Grant (#2020-70006-33000), the NIH Health Research Centers for Minority Serving Institutions Grant (U54MD015959), and USDA Agricultural Research Service through Congress-appropriated funds. The funders had no role in study design, data collection and analysis, decision to publish, or preparation of the manuscript.

**Competing interests:** The authors have declared that no competing interests exist.

## Introduction

Chemistry has a profound influence on every physical system in the human environment [1–5], hence the need for biochemical research is of utmost importance. Gas chromatography coupled with mass spectrometry (GC -MS), used to identify the chemical composition of samples, is a commonly used technology across many disciplines of research [6–9]. While the accuracy and efficiency of instruments continues to improve [10], preparing the library-matched output [i.e. top hit(s) for each set of molecular fragments streamed across the machines $m/z$ detector] for analysis and interpretation remains antiquated. The traditional methods involve manually selecting, integrating, and identifying peaks based on a reference library and comparison to commercial standards across every sample in an experiment [11, 12]. Software that quickly and accurately identify top library matches for every tentative compound in an entire batch of experimental samples thankfully exist (e.g. Agilent's MassHunter, Thermo Fisher Scientific's Compound Discoverer, Shimadzu's GCMSsolution); however, the output remains uninterpretable without additional process. In even simple experiments, the process of quantifying tentatively "identified" compounds across replicates can take weeks or months and is a significant impediment to collecting and analyzing many, large, and/or complex GC -MS datasets. Furthermore, focusing the interpretation on specific chemicals or chemistries that are meaningful would require looking up each molecule for published information and/or important associations. This additional bottleneck in chemical experimentation can lead to backlogs in collections, delays in chemical data being analyzed and published, and may even create a significant deterrent to collecting GC -MS data in studies (e.g. non-targeted and/or suspect screening analysis) where these data could be highly informative.

Another concern with manually selecting component areas for the same tentative molecule across different samples is the inherent subjectivity and inconsistency at many decision points. Every additional keystroke or choice about threshold provides an opportunity for unintended error. Technology exists that could help automate this process, converting the identified compounds into a digitally comparable structure in an instant [4, 8]; however, using such technology requires advanced computer programming experience. Any functional interpretation of a chemical benefits from structural comparisons with compounds of known functions, yet the ability to do so has historically been reserved for private industry or hyper-specialized professionals. A package that automates the sorting and collection of component areas across samples in an experiment while simultaneously storing critical information about every tentative molecule could propel every field of science forward by not only removing the bottlenecks and subjectivity in chemical analysis but also removing the need for hours of paid or untrained manual labor before even simple chemical interpretations can occur.

To address these barriers in the use of GC -MS data, we developed an R package that takes the raw, aggregated chemical identifications generated from a user-selected peak detection software. In this study, we used Agilent's Unknowns Analysis software to identify peaks with their deconvolution algorithm and match $m/z$ spectra to a locally installed NIST library, but any mass spectrometry software that produces the same information is equally viable. The package here comes after the initial processing of samples and communicates with public chemistry utilities (including PubChem and the National Cancer Institute) to sort and process the aggregated set of all tentatively identified molecules using underlying $m/z$ (mass/charge of chemical fragments) ratio data and automatically interpret close matches across samples. In addition to precisely (but flexibly) grabbing tentative compounds from samples they could theoretically exist in and preparing the component areas for statistical summary and analysis–including principal component analyses, non-metric multidimensional scaling (NMDS), and/or machine learning algorithms; *uafR* also interacts with structural data [in SDF (Structure-

Data format)] for all published compounds in the dataset. These data allow detailed summaries of the chemical constituents for each sample to be generated based on the user's chemical(s) of interest. Thus, while a chemical ecologist may be more interested in the relative proportions of alkaloids to polyphenols in a sample [13, 14], a biochemist may only be interested in steroids [6, 15–17]. These groups (or others) can now be selectively pulled from one's dataset to perform follow-up analyses. In addition, researchers (e.g. those performing targeted analysis) that have advanced knowledge of the molecule(s) or functional group(s) of interest can use our functions to isolate these chemistries from experimental data and focus their analysis/interpretation on specified chemicals or chemical groups more generally.

Users may also load personal chemical libraries, again as an easily formatted ".CSV" file using long or wide orientation, to compare any list of chemicals against the set(s) of classifier compounds in their.CSV input library. For the chemical structure processing, our package utilizes Tanimoto similarity, a commonly used and rigorously tested metric for physicochemical comparisons [5, 18, 19]. While there is a broad range of diversity in the chemistry of any system [20], there exists common structural subunits that can categorize molecules into their potential function(s) and the Tanimoto index provides efficient functional sorting of even diverse chemistries. As an example, these comparisons could be used in agricultural research to rapidly screen plant molecules for insecticidal or repellent properties [21–24]. More specifically, the pharmaceutical industry uses the Tanimoto similarity metric to discover compounds that will bind known ligands or share biological activity with known drugs [25–27]. This metric is underapplied, however, because to date, it requires multiple complex steps to generate or acquire data in the appropriate format [28, 29]. Our R package harnesses direct connections between PubChem and R to stream published information on every known (i.e., published and vetted by peer-review for merit) chemical in the dataset. This bypasses the need for other computer programs or coding environments to perform physicochemical comparisons and allows our algorithm to outperform any comparable utility for this stage of mass spectrometry data processing. If the user can install a package and read a ".CSV" file into R, they will have access to the entirety of PubChem and more.

Data science and informatics can circumvent analytical bottlenecks [30]. Automating the tedious portions of GC -MS data processing can not only turn weeks or months of work into a few keyboard strokes within a day, but also takes human error and subjectivity out of the equation. An efficient and user-friendly tool for interpreting these chemical data is long overdue.

Here, we present two examples to demonstrate the accuracy and efficiency of *uafR*. The first is the identification and analysis of a GC/MS dataset containing samples of a series of four known internal standards at different concentrations. The second is a re-identification of GC/MS samples from an already published dataset by Ponce et. al [31]. For this dataset, we compare the same statistical tests for the standardized areas for compounds identified with 4 methods by the *uafR* package and those from the published, manual identifications. We also briefly describe how the package can improve chemical workflows in non-GC -MS datasets or meta-analyses.

## Materials and methods

### Software description and workflow

The current build of *uafR* is optimized for raw output from Agilent's Unknowns Analysis Software (Santa Clara, CA, USA, 95051); however, the only aspect of the workflow that is specific to their software are the column names for the input data frame. To briefly describe the output, after setting up the analysis environment [i.e., directing Unknowns Analysis to the sample directory where a ".UAF" file (hence, *uafR*) is created], running the deconvolution algorithm

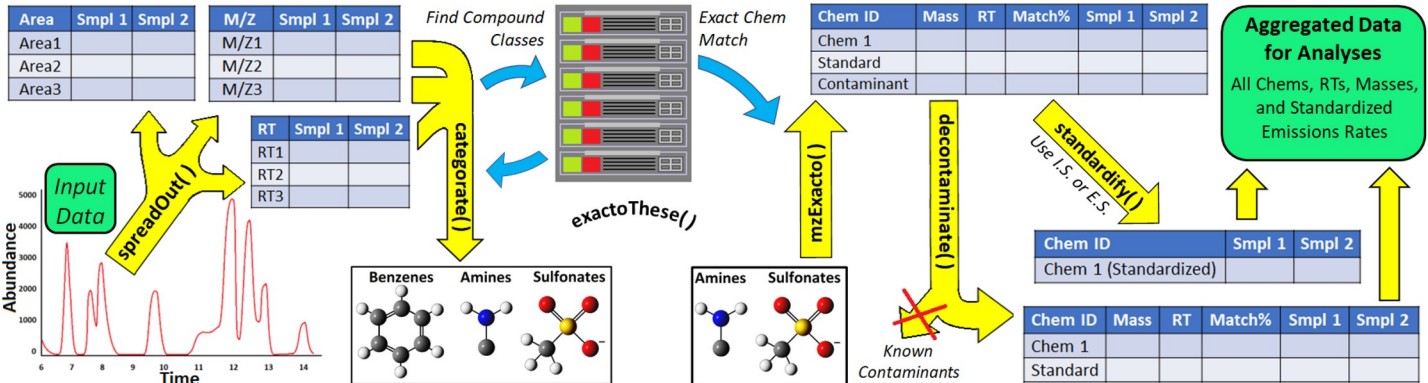

**Fig 1. The *uafR* workflow and its constitutive functions: *spreadOut*, *categorate*, *exactoThese*, *mzExcato*, *decontaminate*, and *standardifyIt*.**

to identify peaks, and searching the peaks against the installed library (blank subtraction and target matching are also options and will not affect the input for *uafR*), a single ".CSV" file containing basic GC-MS output [i.e. retention times, peak area, captured mass-to-charge ratios (*m/z*), compound name, match quality] and a sample origin identifier (i.e. sample name or file name) for tentative compounds across all samples can be exported and read into R using "read.csv()." After reading the data into R and loading the package, *uafR* can use published information to sort and precisely select portions of the data that the user may be interested in.

A diagram of the workflow can be seen in **Fig 1**. The first function for GC-MS data is "*spreadOut*()." Running this function on properly formatted GC-MS input will automatically prepare the data for the next steps in the processing pipeline. Briefly, the function takes every recorded data point for every treatment and expands it in large database formats with unique identifiers assigned for each data point. These unique identifiers (unique IDs) are automatically created from the input data and are used to extract specific area values from the raw data. In addition to setting up large databases containing component area, tentative compound identities, match factors, captured *m/z* values, retention time indices, sample identities, and the unique IDs, the function also communicates with online databases to download relevant information about every tentative compound. To collect these data, the function converts the chemical names into PubChem compound identifiers (CIDs) using the "get_cid()" function from the R package *webchem* [32]. For published chemicals, this information includes exact mass, *m/z* histograms, and every name it has. Instances where the chemical cannot be identified by name on PubChem (i.e. compounds for which a CID are unavailable) are redirected to CADD Group Chemoinformatics Tools and User Services (CACTUS, https://cactus.nci.nih.gov/) from which a canonical Simplified Molecular Input Line Entry System (SMILES) can be generated using that server and algorithm. This SMILES notation is then used to simulate the mass and structure data for, as-of-yet unpublished chemicals on PubChem. All this information, including the large databases, are stored as a list in a user-defined object. Subsequent functions are designed to seamlessly interact with the list and will automatically use relevant information collected during "*spreadOut*()".

The next step in the GC-MS workflow will depend on the type of analysis the user is performing. If the chemicals of interest are already known, they can be extracted by name with a single function—"*mzExacto*()." However, for complex datasets or analyses that involve more unknowns, the user may want to cast a broader, but still accurate, net. There are multiple steps that can be taken to hone in on the most relevant chemicals in a dataset using the features of *uafR*. A simple and effective approach is to subset the search chemicals by setting a minimal

match factor on the raw output of Unknowns Analysis (or other GC-MS software). This can be done with R code described in the vignette published with the package (https://castratton. github.io/uafR/). Another approach could include subsetting with output from the function "*categorate*()." This function also uses PubChem to communicate with online databases and generate categorically, structurally, and chemically identifying information for every published chemical in the dataset. The categorical data include whether the chemical is biologically derived [Natural Products Online database (LOTUS; https://lotus.naturalproducts.net/)], has flavor or smell [Flavor and Extract Manufacturers Association (FEMA; https://www. femaflavor.org/)], has varied biological activities [Kyoto Encyclopedia of Genes and Genomes (KEGG; https://www.genome.jp/kegg/)], medical subject headings (MeSH; https://www.nlm. nih.gov/mesh/), or other information about their reactivity [Food and Drug Administration—Structured Product Labeling (FDA/SPL; https://www.fda.gov/) and Reactive Groups from PubChem (https://pubchem.ncbi.nlm.nih.gov/)].

After the categorical information is collected, the function generates substructure data for the chemicals to also be subsetted by common functional groups. This information is generated using the "read.SDFset()" function from another R package called *ChemmineR* [33]. This package is a dependency that is installed with *uafR* and is core to the cheminformatics methods deployed. The substructure information generated using *ChemmineR* includes the number of rings, all subgroups (e.g., R-COH, R-COOH, etc.) and their counts, all atoms (e.g., C, N, S, As, etc.) and their counts, and the number of charges for every chemical with published structural data (or canonical SMILES from CACTUS) on PubChem. The final steps in "*categorate*()" will not only assist in subsetting compounds of interest for extracting from GC-MS datasets, but could also be used to perform meta-analyses on published chemistries.

In order to run "*categorate*()," users are required to include an input library that contains columns with labeled chemicals. The labels are customizable, but the most useful approach is to label a set of chemicals by a common feature or biological activity. For example, if a researcher has a set of plant chemicals of interest to test against active ingredients in pharmaceuticals, the input library could contain *n* columns whose headings are the biological activity (e.g., diuretic, blood pressure, etc.) and the contents (rows under the heading) are the active chemicals used in products that are approved for those medical outcomes. The "*categorate*()" function will then take the input library (saved as a ".CSV") and compare every chemical of interest to the chemicals in each user-defined "chemical category," returning two additional data frames—(1) whether it has a strong (Tanimoto similarity greater than 0.95) or moderate (greater than 0.85) structural match with any of the chemicals in each group; and, (2) for strong matches, the name of a chemical that it was most similar to. It performs these comparisons using the "fmcsBatch()" function from the R package *fmcsR* [34].

The utility of this information and approach cannot be overstated. For chemistry, structure defines function, so identifying structural matches is effectively identifying chemicals with the same function. This not only provides a powerful tool for novel chemical activity discoveries and/or natural backups to synthesized chemistries, but can also allow researchers to subset GC-MS data by general chemical structures or activities they are interested in. The possibilities are limited only by the maximum file size a user can create in the specified ".CSV" format and whether structural data were able to be generated from PubChem for the chemical(s). Subsetting of information generated with "*categorate*()" is easily done using the function–"*exactoThese*()." Users can specify which set of information they would like to subset and indicate desired criteria the chemicals should meet.

Next in the GC-MS workflow is to put the published information to use and aggregate every occurrence of the user-specified chemicals across every GC-MS sample. "*mzExacto*()" takes the output from "*spreadOut*()" along with the list of chemical names, and returns a single

data frame containing their optimal retention time, exact mass, best identified match factor, and aggregated component area across samples in which it occurs (0 when absent). Additional technical details for this algorithm are available with the package (github.com/castratton/uafR). Briefly, after collecting mass and *m/z* information for the input chemicals of interest, they are ordered by exact masses so likely retention time windows can be determined based on the general structure of the input data and the information stored from "*spreadOut*()." After identifying perfect matches (i.e., those with high match factors and the same chemical names) the algorithm looks again through each sample for instances where the top 2 published *m/z* values for the tentative identity are the same as the query chemicals of interest. These matches are based on standard manual approaches to resolve uncertainties in any complex GC/MS workflow. The *m/z* values within retention time windows generated by the input data must be similar enough that the chemical fragments are practically and theoretically identical. A sub-argument, "decontaminate," is on by default and removes any chemicals that did not have a strong match across samples, were unable to be found in public databases (i.e. PubChem), and/or were unable to have a canonical SMILES generated on the NCI server. This sub-argument can be turned off by adding "decontaminate = F" to the end of the items in "*mzExacto*()."

At this point in the GC/MS workflow, the most common step is to standardize component areas for tentatively identified chemicals by quantifying their values relative to known internal or external standard(s). "*standardifyIt*()" takes the output from "*mzExacto*()" and either a user-specified internal standard (e.g., tetradecane, or user defined-internal standard) or calibration curves (raw values) from an external standard(s), along with sub-arguments that allow the standardization to be tuned to the experimental methods. "*standardifyIt*()" returns a data frame that is standardized relative to the known chemical quantifications and formatted for subsequent statistical analyses. Common statistical protocols for GC -MS data include ordination analyses (e.g., PCA, NMDS, etc.), multivariate statistical tests (e.g., ANOSIM, MANOVA, PERMANOVA, etc.) and/or deep learning (neural networks or machine learning). Each of the required formats for running these statistics on GC -MS data are achievable with the final output of "*mzExacto*()" and "*standardifyIt*()."

Beyond automating a process that can require hours of work per sample, with potentially hundreds of samples per study, *uafR* makes cheminformatics a possibility for anyone working with GC -MS or chemical identity data. Furthermore, the public databases our package accesses will only improve in data quality/quantity with time and increased use. To showcase the utility and validity of our package for GC -MS workflows, we analyzed two datasets–one containing a set of known standards pipetted in known quantities across three samples (low, medium, and high concentrations) and the other, consisting of a recently published set of 35 samples.

## Testing the accuracy of the package

The first dataset on which *uafR* functions were tested was a series of standards including, ethyl hexanoate (Prod#14896, CAS#123-66-0, Millipore Sigma, Burlington, MA, USA), methyl salicylate (Prod#M6752, CAS#119-36-8, Millipore Sigma), octanal (Prod#S7303001712, CAS#124-13-0, Merck, Darmstadt, Germany), and undecane (Prod#S7466429734, CAS#1120-21-4, Merck, Darmstadt, Germany) collected on an Agilent 7890b gas chromatograph (GC) equipped with an Agilent Durabond HP-5 column (30 m length, 0.250 mm diameter, and 0.25 µm flm thickness) using He as the carrier gas at a constant 5 ml/min flow and 39 cm/s velocity, coupled with an Agilent 5997B mass spectrometer (MS) single quadrupole detector. These were prepared in a serial dilution using 1 mL of neat compound from each and diluting in 10 mL in dichloromethane, and subsequently moving 1 mL of the dilutant to a new

container with 10 mL of fresh dichloromethane until the following amounts of ethyl hexano-ate, methyl salicylate, octanal, and undecane were achieved: low (0.000087 ng, 0.0001179 ng, 0.000082 ng, and 0.000072 ng, respectively), medium (434.5 ng, 589.5 ng, 410 ng, and 370 ng, respectively), or high (2172.5, 2947.5 ng, 2050 ng, and 1850 ng, respectively) relative quantities.

The second dataset was GC/MS data collected on the same GC-MS as the previous test chemicals that had been manually processed and published in August, 2022. Briefly, the samples were collected from grain samples that were a) UV sterilized (negative control), b) clean grain from storage (positive control), c) inoculated with asexual fungal spores, d) inoculated with sexual fungal spores (see Ponce et al. 2022 [31] for extended description of methods).

After analyzing the samples on the GC-MS, the raw output is saved to a local directory and loaded into Unknowns Analysis following default protocols. For a detailed overview of running this software, Agilent provides a user manual. After loading the samples and loading the methods file to every sample, the deconvolution algorithm identified the most accurate peaks for every chromatogram. Each peak was then searched against the NIST 20 database. The aggregated data frame was exported as a ".CSV" file. This data frame included columns for the compound names ("Compound.Name"), file name the tentative identity is from ("File. Name"), top *m/z* peaks captured by the GC/MS ("Base.Peak.MZ"), match factors for tentative identities ("Match.Factor"), and retention times ("Component.RT").

## Results

Peak areas calculated by *uafR* for the set of standards correlated with the volume of the standards injected, with $R^2$ values ranging from 0.8273 to 0.9998 (**Fig 2**). Importantly, the single standard (e.g., octanal) with a lower correlation coefficient was likely misread by the MS or had volatized prior to being run on the GC-MS. It is known that that octanal volatilizes very easily, and is used by plants as an anti-fungal compound to protect fruit [35].

After confirming that *uafR* can precisely identify chemicals that are known to be in a sample, the next step was to assess its accuracy in a more complex experiment with unknowns. Using raw GC-MS data from a recently published experiment allowed the workflow to be tested against a peer-reviewed study. We found that *uafR* was able to identify the manually selected compounds with accurate matches to manually identified retention times (**Table 1**) and yielded the same overall pattern of significance in ANOSIM analysis (**Table 2**). The true benefit of using *uafR* is not merely its accuracy, but also its speed. For context, the original manual identifications required months of labor. Using *uafR*, we re-analyze this entire experiment in 150 minutes of automated computation using a standard desktop computer with a 3.30 GHz processor and 16 GB RAM. While the speed and accuracy for this experiment are apparent, additional trials on larger datasets are warranted.

The possible applications of a direct connection between R and PubChem are diverse. Beyond statistical tests and advanced computational pipelines, the graphical framework can provide publication quality visuals with minimal code. This package harnesses the most advanced open-source chemical dataset and makes it accessible to anyone with basic experience working in R.

## Conclusion

Our described workflow and package utilities bring GC -MS data processing up to par with the advanced technology that generates the data. Though technically uafR should apply in the same manner to other mass spectrometers (e.g. Q-TOF, Orbitrap, TimsTOF HT, and/or Astral) and even liquid chromatography coupled MS data, it has not yet been tested in those

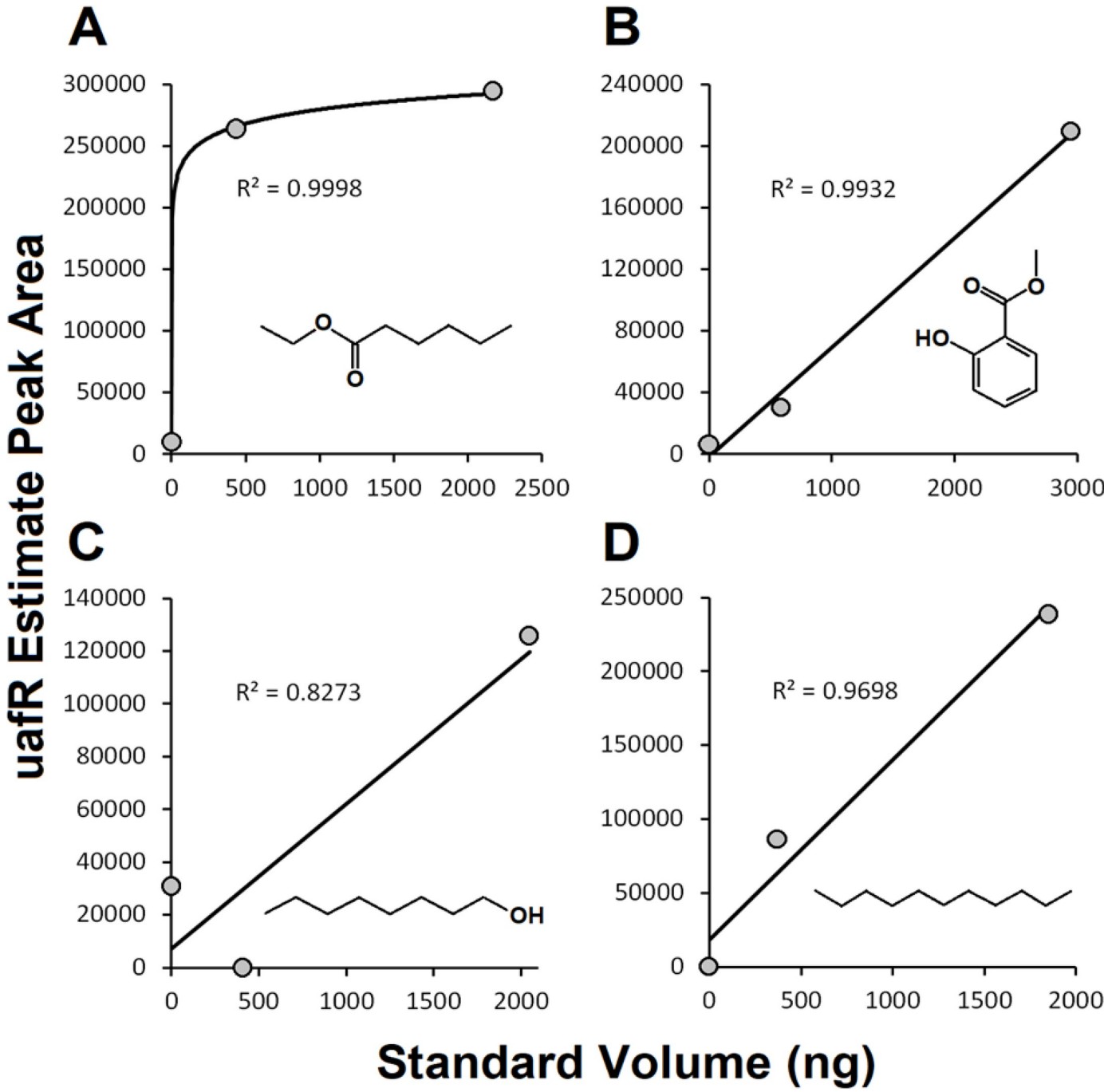

**Fig 2.** Correlations between volume of standards tests via GC/MS and the peak area estimates generated by uafR, for (A) Ethyl hexanoate, (B) Methyl salicylate, (C) Octanal, (D) Undecane. Points represent raw data while the line represents a natural log fit (A) or linear fit (B-D) to the raw data.

contexts. In addition, since uafR depends on published data, our algorithms do not yet apply to MS2, MS3 or ion mobility + MS2 data. This functionality will be added to the software once these spectra are published for most chemicals in PubChem's database. The difficult portion of chemical identifications should not occur on the computer. Anyone with the ability to install packages and load a ".CSV" file into R now has access to a suite of functions that streamline a

**Table 1. Summary of NMDS and ANOSIM calculation for models processed with *uafR*.**

| Model | NMDS Stress | ANOSIM | | # Compounds in Final Table |
|---|---|---|---|---|
| | | *R* | *P* | |
| Original Ponce et al. 2022 | 0.10 | 0.20 | 0.001 | 33 |
| uafR Ponce et al. 2022 | 0.11 | 0.185 | 0.009 | 33[a] |
| >65% Match Factor | 0.17 | 0.068 | 0.17 | 427 |
| >75% Match Factor | 0.11 | 0.016 | 0.33 | 116 |
| >88.9 Match Factor | 0.15 | 0.034 | 0.29 | 30 |
| >97.2 Match Factor | 0.04 | -0.034 | 0.58 | 3 |

[a] Only 30 compounds used for analysis, since three compounds were not present in enough experimental replicates

**Table 2. Chemicals identified in Ponce et al. 2022 using manual identification, versus compounds identified by the *uafR* package using the same selection criteria: >75% match of the chemical ID, and present in more than one sample.** Compounds shared between identification techniques are in bold print.

| Ponce et al. 2022 | | uafR | |
|---|---|---|---|
| Chemical ID | RT | Chemical ID | RT |
| Pivalaldehyde, semicarbazone | 4.735 | | |
| 2-Butenal | 4.788 | | |
| **2,4-Dimethyl-1-heptene** | **4.792** | **2,4-Dimethyl-1-heptene** | **4.796** |
| **2-Pentanone, 4-hydroxy-4-methyl** | **4.802** | **2-Pentanone, 4-hydroxy-4-methyl-** | **4.804** |
| | | Cyclopentanone, 2-methyl- | 5.513 |
| Benzene, propyl | 6.292 | | |
| | | Benzene, 1-ethyl-3-methyl- | 6.404 |
| 1-Octen-3-ol | 6.516 | | |
| Butanal | 6.525 | | |
| 4,5-Dichloro-1,3-dioxolan-2-one | 6.628 | Benzene, 1-ethyl-4-methyl- | 6.633 |
| **3-Octanone** | **6.648** | **3-Octanone** | **6.645** |
| Decane | 6.823 | | |
| **Mesitylene** | **6.869** | **Mesitylene** | **6.872** |
| **Benzene, 1,2,4-trimethyl-** | **7.319** | **Benzene, 1,2,4-trimethyl-** | **7.017** |
| D-Limonene | 7.364 | | |
| Benzeneethanol, beta-ethyl | 7.541 | | |
| **Benzene, 1,4-diethyl** | **7.659** | **Benzene, 1,2-diethyl-** | **7.418** |
| **Benzene, 1,2-diethyl** | **7.759** | **Benzene, 1,4-diethyl-** | **7.738** |
| 1,3,8-p-Menthatriene | 7.86 | Limonene | 7.780 |
| | | p-Cymene | 8.200 |
| | | Benzene, 2-ethyl-1,4-dimethyl- | 8.203 |
| 4-Dichloromethyl-2[[2-[1-methyl-2-pyrrolidinyl]ethyl]amino-6-Trichloromethylpyrimidine | 8.271 | | |
| Benzene, (2-methyl-1-propenyl)- | 8.279 | | |
| 1-Phenyl-1-butene | 8.282 | | |
| **Linalool** | **8.314** | **Linalool** | **8.315** |
| | | Undecane | 8.329 |
| **Nonanal** | **8.37** | **Nonanal** | **8.373** |
| 2-Thiophenecarboxylic acid, 5-nonyl- | 9.044 | Cyclopentasiloxane, decamethyl- | 8.950 |
| Dichloroacetaldehyde | 9.735 | | |
| | | Cyclopentanecarboxylic acid, pentyl ester | 10.185 |
| Linalyl acetate | 10.558 | 1,6-Octadien-3-ol, 3,7-dimethyl-, formate | 10.559 |

*(Continued)*

**Table 2.** (Continued)

| Ponce et al. 2022 | | uafR | |
| --- | --- | --- | --- |
| **Chemical ID** | **RT** | **Chemical ID** | **RT** |
| Beta-Ocimene | 10.561 | | |
| 2-Thiophenecarboxylic acid | 10.59 | 2-Thiophenecarboxylic acid, 3-methylbutyl ester | 10.590 |
| 1-Pent-3-ynylcyclopenta-1,3-diene | 10.653 | | |
| | | Ethanone, 1-(2,5-dimethylphenyl)- | 10.816 |
| 1,5,6,7-Tetramethylbicyclo[3.2.0]hepta-2,6-diene | 10.822 | Ethanone, 1-(3,4-dimethylphenyl)- | 10.821 |
| | | Ethanone, 1-(2,4-dimethylphenyl)- | 10.916 |
| **Ethanone, 1-(4-ethylphenyl)** | **11.108** | **Ethanone, 1-(4-ethylphenyl)-** | **10.973** |
| | | Cyclotetrasiloxane, octamethyl- | 13.749 |
| Butyl citrate | 21.812 | | |
| 1-Methyl-4-phenyl-5-Thioxo-1,2,4-triazolidin-3-one | 23.957 | | |
| 9-Octadecenamide, (Z-) | 26.311 | | |

complex workflow so more effort can be spent interpreting rather than preparing data. It is important to mention that while uafR accurately processed the GC/MS data tested here, researchers should still validate that the compound areas identified by the algorithm make chemical and/or biological sense in their study system. Thankfully, the output from categorate () can help in these assessments by collecting relevant information for every molecule in an easily interpretable structure.

Chemical knowledge has grown increasingly advanced and accessible in recent years. The precision of GC -MS instruments and, consequently, their output, allows published information to be accessed with 100% accuracy. While previous algorithms have focused on using statistics to separate likely aggregates of compound areas, their accuracy fails in complex contexts because too many distinctly different chemicals "behave" (i.e., have the same mass and/or retention indices) the same so cannot be teased out statistically without additional knowledge.

Our approach is the first and, to date, only R package that uses published data to extract compound areas for the most likely compound identifications. By automating this component of the GC -MS workflow, we anticipate our package will greatly increase the speed at which chemistry datasets are published, the size of chemical studies that can be conducted, and the accessibility of chemical analyses to scientists in related fields.

## Acknowledgments

The use of trade names is for the purposes of providing scientific information only and does not constitute endorsement by the United States Department of Agriculture. The USDA is an equal opportunity employer.

## Author Contributions

**Conceptualization:** Chase A. Stratton, Yvonne Thompson.

**Data curation:** Chase A. Stratton, Yvonne Thompson.

**Formal analysis:** William R. Morrison, III, Ebony G. Murrell.

**Funding acquisition:** Chase A. Stratton, William R. Morrison, III, Ebony G. Murrell.

**Investigation:** Chase A. Stratton, Yvonne Thompson.

**Methodology:** Chase A. Stratton, Yvonne Thompson, Konilo Zio.

**Project administration:** Chase A. Stratton.

**Resources:** William R. Morrison, III.

**Software:** Chase A. Stratton, Yvonne Thompson, Konilo Zio.

**Validation:** Chase A. Stratton, Yvonne Thompson, Konilo Zio, William R. Morrison, III, Ebony G. Murrell.

**Visualization:** Konilo Zio.

**Writing – original draft:** Chase A. Stratton, Yvonne Thompson, William R. Morrison, III, Ebony G. Murrell.

**Writing – review & editing:** Chase A. Stratton, Konilo Zio, William R. Morrison, III, Ebony G. Murrell.

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
