## [Decision Letter · Decision Letter 0]

11 Apr 2024

PONE-D-24-10543uafR: An R package that automates mass spectrometry data processingPLOS ONE

Dear Dr. Murrell,

Thank you for submitting your manuscript to PLOS ONE. After careful consideration, we feel that it has merit but does not fully meet PLOS ONE’s publication criteria as it currently stands. Therefore, we invite you to submit a revised version of the manuscript that addresses the points raised during the review process.

We look forward to receiving your revised manuscript.

Kind regards,

Shailender Kumar Verma, Ph.D.

Academic Editor

PLOS ONE

 [This project was funded by USDA-NIFA projects: #2021-67034-35135 and #2018-67013-27402; and by generous private donations to The Land Institute. In addition, this work was funded, in part, by a United States Department of Agriculture, National Institute of Food and Agriculture, Crop Protection and Pest Management Grant (#2020-70006-33000) and USDA Agricultural Research Service through Congress-appropriated funds.].  

[The use of trade names is for the purposes of providing scientific information only and does not constitute endorsement by the United States Department of Agriculture. The USDA is an equal opportunity employer.  This project was funded by USDA-NIFA projects: #2021-67034-35135 and #2018-67013-27402; and by generous private donations to The Land Institute. In addition, this work was funded, in part, by a United States Department of Agriculture, National Institute of Food and Agriculture, Crop Protection and Pest Management Grant (#2020-70006-33000) and USDA Agricultural Research Service through Congress-appropriated funds.]

 [This project was funded by USDA-NIFA projects: #2021-67034-35135 and #2018-67013-27402; and by generous private donations to The Land Institute. In addition, this work was funded, in part, by a United States Department of Agriculture, National Institute of Food and Agriculture, Crop Protection and Pest Management Grant (#2020-70006-33000) and USDA Agricultural Research Service through Congress-appropriated funds.]

Reviewers' comments:

Reviewer's Responses to Questions

**Comments to the Author**

1. Is the manuscript technically sound, and do the data support the conclusions?

Reviewer #1: No

Reviewer #2: Yes

Reviewer #3: Yes

Reviewer #4: Yes

Reviewer #5: Yes

Reviewer #6: Yes

2. Has the statistical analysis been performed appropriately and rigorously? 

Reviewer #1: No

Reviewer #2: Yes

Reviewer #3: Yes

Reviewer #4: Yes

Reviewer #5: Yes

Reviewer #6: N/A

3. Have the authors made all data underlying the findings in their manuscript fully available?

Reviewer #1: Yes

Reviewer #2: Yes

Reviewer #3: Yes

Reviewer #4: Yes

Reviewer #5: Yes

Reviewer #6: Yes

4. Is the manuscript presented in an intelligible fashion and written in standard English?

Reviewer #1: Yes

Reviewer #2: Yes

Reviewer #3: Yes

Reviewer #4: Yes

Reviewer #5: Yes

Reviewer #6: Yes

5. Review Comments to the Author

Reviewer #1: The authors presented a manuscript that describes the creation of an R package to process Raw GC-MS data. They also claim applicability to LC-MS data, however failed to present any data to support nor addressed the fundamental difference between GC-MS and LC-MS data that would make the work flow a challenge. The authors compare their process to “manually selecting, integrating and identifying peaks” that fails to compare to any open source or commercial solution. For a non-targeted experiments, it would be rare for a lab not to use some level of processing software. The manuscript would be improved greatly by describing the current GC-MS solution and describing how their code improves on their shortcomings. The manuscript was not clear on the process of annotation. Was the EI spectra search against NIST library? Was there accurate mass searching used? Or both? If the later, how was a parent mass obtained? What was the resolution and mass accuracy of the instrument used to collect the dataset described?

The author claim “the most accurate compound areas”, however little support to support this claim. This would be to be compared to other peak detection algorithms and not manual process which the author stated is full of bias. In addition, any comparison to speed needs comparison to other algorithms instead of manual process. Th size of the datasets used it the examples were relatively small and do not support any claim to ability to handle large datasets.

The authors presented a process that was built from what appears to be existing code to perform the tasks numerous software packages can do. There is no evidence to that this process would be applicable to LC-MS data.

Consistently use GC-MS, and m/z should be italicized.

Reviewer #2: In this manuscript entitled “uafR: An R package that automates mass spectrometry data processing”, the authors were trying to demonstrate the workflow and R package uses published data to extract the most accurate compound areas for the most likely compound identifications, and will greatly increase the speed at which chemistry datasets are published, the size of chemical studies that can be conducted, and the accessibility of chemical analyses. This reviewer believes this manuscript will be beneficial for readers of PLOS One to some extent. However, the authors need to address following concerns before could be considered to publish on PLOS ONE.

Comments and Concerns:

1. In the manuscript, the authors did not mention which mass spectrometer was used for the first dataset, only demonstrate the second dataset of GC-MS data from Agilent 5997B mass spectrometer, which is single quadrupole detector. The authors should provide more examples or demonstrations on other mass spectrometers from different vendors or other types of mass spectrometers, like Q-TOF or Orbitrap, even most cutting-edge ones like TimsTOF HT or Astral.

2. If uafR compared with commercial softwares, like Compound Discoverer from Thermo, what are the advantages of uafR compared to these commercial softwares?

3. It will be great if the authors could address whether the uafR could handle MS2 or MS3 data, additionally whether uafR could handle ion mobility + MS2 data.

Reviewer #3: The authors have developed a new tool for processing large volumes of GC-LC/MS data in a minimum of time. They validated this tool using 2 different data sets. In addition, they allow everyone to test this tool with other data sets. In addition, the methods applied are well described and rigorous.

Reviewer #4: I want to congratulate you on your work. As an LC-MS user, although not a direct part of the target audience, I still appreciate your work very much. I wrote some suggestions in the attached document, which I think would improve the user and reader experience of your target audience. I did find a few small mispellings and typos, please do a careful read and check/correct these.

Reviewer #5: Review for UafR: An open-source R package that automates mass spectrometry data processing

I have previously reviewed this manuscript for a different journal and during the multiple revisions reviewed concluded that the manuscript was ready for publication. Only minor changes have been made since, so I can still recommend this for publication as is. For the sake of transparency and thoroughness, I've copied my prior review pertaining to this manuscript below so that the editor can see what was pointed out then and how edits were made. The only comment I'll make here is that the authors could tone down a bit of language used for more succinct and detail-oriented descriptions of implementation and results (for example, lines 235-237). Though I'll acknowledge this is personal preference.

Review from April 2023, Journal of Cheminformatics

The authors describe ‘uafR’, an R package to support mass spectrometry data processing. I believe the package the authors have created can provide utility for mass spec users, specifically for those interested in compiling metadata to support annotations. However, I believe the manuscript could be improved with more context and perspective. As is, it is not abundantly clear why a user would elect to use this tool over other similar tools or web resources available. Additionally, it’s difficult to determine what the primary goals of the package are- standard curve generation and metadata/substructure calculation are quite different pieces of a workflow, so describing the primary benefit would make the manuscript clearer as well. I’ve provided section specific comments below.

Introduction

Lines 34-39: here you outline the first impediment, when I see how the rest of the paper is outlined it appears that this step here is achieved with the Unknowns Analysis tool right? So first identifications occur in a tool of choice via library search, etc. and then use this the ‘uafR’ tool for enhanced metadata? I think providing some connection point between the impediments outlined in the introduction and how the implementation and results solve them will be helpful for readers.

Line 56-60: related to the above comment and this section, perhaps what is missing to me is exactly what is included in the outputs of Unknowns Analysis and then what is really being enhanced by the ‘uafR’ tool. Can you provide example outputs? Or snippets via screenshot to help orient the reader?

Implementation

Lines 120-123: So all identifications come from Unknowns Analysis? What if a user doesn’t use Unknowns Analysis?

Line 153-155: This function seems incredibly helpful. I’d recommend highlighting this even further

Line 206-212: How is the standard curve generated here different (or better?) than using the standard curve from elsewhere? Is the standard curve part of Unknowns Analysis too or it needs to be generated here because it is not in Unknowns Analysis? A little more clarity here would help.

Line 247-250: I think I missed a step in here. Are the peak areas calculated in Unknowns Analysis and then compared in ‘uafR’ or are they re-generated in ‘uafR’? Suggest adding a bit more detail around the steps followed.

Reviewer #6: I read the submitted manuscript to try providing a correct report on it.

However, I realized that the paper is presenting an original software able to analyze and report GC-MS and LC-MS data…

I am totally not able to understand how efficient is this software since I am totally not expert in computer script coding…

I can only state that there is a need for such informatic tools but I cannot state if this one is efficient or not.

6. PLOS authors have the option to publish the peer review history of their article (what does this mean?). If published, this will include your full peer review and any attached files.

Reviewer #1: No

Reviewer #2: No

Reviewer #3: No

Reviewer #4: No

Reviewer #5: No

Reviewer #6: No

---

## [Author Response · Author response to Decision Letter 0]

24 May 2024

Please see the attached file, "Rebuttal Letter_uafR PLoS One 2024" for detailed responses to reviewer comments.

---

## [Decision Letter · Decision Letter 1]

13 Jun 2024

uafR: An R package that automates mass spectrometry data processing

PONE-D-24-10543R1

Dear Dr. Murrell,

We’re pleased to inform you that your manuscript has been judged scientifically suitable for publication and will be formally accepted for publication once it meets all outstanding technical requirements.

Kind regards,

Shailender Kumar Verma, Ph.D.

Academic Editor

PLOS ONE

Additional Editor Comments (optional):

Reviewers' comments:

Reviewer's Responses to Questions

**Comments to the Author**

1. If the authors have adequately addressed your comments raised in a previous round of review and you feel that this manuscript is now acceptable for publication, you may indicate that here to bypass the “Comments to the Author” section, enter your conflict of interest statement in the “Confidential to Editor” section, and submit your "Accept" recommendation.

Reviewer #2: All comments have been addressed

Reviewer #4: All comments have been addressed

2. Is the manuscript technically sound, and do the data support the conclusions?

Reviewer #2: Yes

Reviewer #4: Yes

3. Has the statistical analysis been performed appropriately and rigorously? 

Reviewer #2: N/A

Reviewer #4: Yes

4. Have the authors made all data underlying the findings in their manuscript fully available?

Reviewer #2: Yes

Reviewer #4: Yes

5. Is the manuscript presented in an intelligible fashion and written in standard English?

Reviewer #2: Yes

Reviewer #4: Yes

6. Review Comments to the Author

Reviewer #2: (No Response)

Reviewer #4: Congratulations on your hard work! The revised manuscript added some crucial data, and although more could have been said/written in the discussions section of the manuscript, I think your manuscript can be published in its current form. I hope the software package you developed will be freeware, as it would help many analysts who are beginners and/or are struggling with mass spectra interpretations. Best regards,

7. PLOS authors have the option to publish the peer review history of their article (what does this mean?). If published, this will include your full peer review and any attached files.

Reviewer #2: **Yes: **Guanghui Han

Reviewer #4: No

---

## [Editor Report · Acceptance letter]

25 Jun 2024

PONE-D-24-10543R1 

PLOS ONE

Dear Dr. Murrell, 

I'm pleased to inform you that your manuscript has been deemed suitable for publication in PLOS ONE. Congratulations! Your manuscript is now being handed over to our production team.

Kind regards, 

on behalf of

Dr. Shailender Kumar Verma 

Academic Editor

PLOS ONE